# Gabapentin and Duloxetine Prevent Oxaliplatin- and Paclitaxel-Induced Peripheral Neuropathy by Inhibiting Extracellular Signal-Regulated Kinase 1/2 (ERK1/2) Phosphorylation in Spinal Cords of Mice

**DOI:** 10.3390/ph14010030

**Published:** 2020-12-31

**Authors:** Natsuki Kato, Keisuke Tateishi, Masanobu Tsubaki, Tomoya Takeda, Mikihiro Matsumoto, Katsumasa Tsurushima, Toshihiko Ishizaka, Shozo Nishida

**Affiliations:** 1Division of Pharmacotherapy, Kindai University Faculty of Pharmacy, Kowakae, Higashi-Osaka 577-8502, Japan; katou_kindai@yahoo.co.jp (N.K.); tateishi_kindai@yahoo.co.jp (K.T.); tsubaki@phar.kindai.ac.jp (M.T.); takeda@phar.kindai.ac.jp (T.T.); matsumoto_kindai@aol.com (M.M.); tsurushima_kindai@yahoo.co.jp (K.T.); 2Department of Pharmacy, Sakai City Medical Center, Sakai 593-8304, Japan; ishizaka_sakai@yahoo.co.jp

**Keywords:** chemotherapy-induced peripheral neuropathy, gabapentin, duloxetine, ERK1/2, oxaliplatin, paclitaxel

## Abstract

Chemotherapy-induced peripheral neuropathy is a common factor in limiting therapy which can result in therapy cessation or dose reduction. Gabapentin, a calcium channel inhibitor, and duloxetine, a serotonin noradrenaline reuptake inhibitor, are used to treat a variety of pain conditions such as chronic low back pain, postherpetic neuralgia, and diabetic neuropathy. It has been reported that administration of gabapentin suppressed oxaliplatin- and paclitaxel-induced mechanical hyperalgesia in rats. Moreover, duloxetine has been shown to suppress oxaliplatin-induced cold allodynia in rats. However, the mechanisms by which these drugs prevent oxaliplatin- and paclitaxel-induced neuropathy remain unknown. Behavioral assays were performed using cold plate and the von Frey test. The expression levels of proteins were examined using western blot analysis. In this study, we investigated the mechanisms by which gabapentin and duloxetine prevent oxaliplatin- and paclitaxel-induced neuropathy in mice. We found that gabapentin and duloxetine prevented the development of oxaliplatin- and paclitaxel-induced cold and mechanical allodynia. In addition, our results revealed that gabapentin and duloxetine suppressed extracellular signal-regulated protein kinase 1/2 (ERK1/2) phosphorylation in the spinal cord of mice. Moreover, PD0325901 prevented the development of oxaliplatin- and paclitaxel-induced neuropathic-like pain behavior by inhibiting ERK1/2 activation in the spinal cord of mice. In summary, our findings suggest that gabapentin, duloxetine, and PD0325901 prevent the development of oxaliplatin- and paclitaxel-induced neuropathic-like pain behavior by inhibiting ERK1/2 phosphorylation in mice. Therefore, inhibiting ERK1/2 phosphorylation could be an effective preventive strategy against oxaliplatin- and paclitaxel-induced neuropathy.

## 1. Introduction

Chemotherapy-induced peripheral neuropathy (CIPN) is a common dose-limiting adverse effect of multiple chemotherapeutic agents, including oxaliplatin and paclitaxel, which can result in therapy cessation or dose reduction [1,2,3]. In addition, neuropathy causes a significant loss of functional abilities and decreases the quality of life of patients. Among chemotherapeutic agents, platinum drugs (oxaliplatin and cisplatin), vinca alkaloids (vincristine and vinblastine), taxanes (paclitaxel and docetaxel), and bortezomib induce the most severe effects on the peripheral nervous system [4,5,6,7]. Since little is known about the mechanisms involved in the development of CIPN, no drugs are currently available to prevent or treat CIPN. Thus, the identification of drugs that effectively prevent or treat CIPN is important.

The activation of protein kinase C (PKC)/extracellular signal-regulated kinase 1/2 (ERK1/2) pathway contributes to the development of neuropathic pain. Administration of oxaliplatin to rats evoked thermal and mechanical hypersensitivity via increased PKC/ERK1/2 activation in the spinal cord and cortical areas [8,9,10]. Treatment with paclitaxel induced mechanical hypersensitivity via increased ERK1/2 activation in the dorsal root ganglion (DRG) of rats [11]. In addition, PKC-induced ERK1/2 phosphorylation has been associated with neuropathic pain in rats [12,13]. Moreover, it has been reported that PKC mediates a positive feedback mechanism regulating calcium channel activity, which may affect nerve sensitivity during chemotherapy [14,15].

Gabapentin, an anticonvulsant, has been used to treat a variety of non-epileptic conditions such as chronic pain, psychiatric disorders, and movement disorders. It is thought to act by binding to the α2δ subunit of voltage-dependent calcium channels, thus inhibiting the release of excitatory neurotransmitters [16,17]. Duloxetine, a serotonin-noradrenaline reuptake inhibitor, is used to treat painful diabetic neuropathy, osteoarthritis-related pain, and chronic low back pain [18,19,20,21]. A number of studies have reported that these drugs suppress oxaliplatin- and paclitaxel-induced mechanical hyperalgesia and cold allodynia in rat and mice models [22,23,24]. However, the mechanisms underlying this analgesic effect remains unclear. Therefore, the present study investigated the mechanism by which gabapentin and duloxetine suppress oxaliplatin- and paclitaxel-induced neuropathic pain.

## 2. Results

### 2.1. Effects of Gabapentin on Oxaliplatin- and Paclitaxel-Induced Cold and Mechanical Allodynia

To evaluate the effect of gabapentin on oxaliplatin- and paclitaxel-induced cold sensitivity, mice received oxaliplatin or paclitaxel on days 0 and 7 and were administered gabapentin (30 or 100 mg/kg/day, p.o.) daily. Oxaliplatin and paclitaxel induced a significant progressive reduction in withdrawal thresholds at 10 °C. Oral administration of gabapentin prevented the development of oxaliplatin- and paclitaxel-induced cold allodynia (Figure 1A and Figure 2A). No significant differences were observed in withdrawal latency at any time point by the combination treatment with 30 mg/kg gabapentin and 6 mg/kg oxaliplatin or 6 mg/kg paclitaxel compared with the combination treatment with 100 mg/kg gabapentin and 6 mg/kg oxaliplatin or 6 mg/kg paclitaxel (Figure 1A and Figure 2A). In addition, we investigated the effect of gabapentin on the development of oxaliplatin- and paclitaxel-induced mechanical allodynia using the following von Frey filaments: 0.16 g (mechanical allodynia), 0.4 g (intermediate), and 1.4 g (mechanical hyperalgesia). Administration of 30 mg/kg gabapentin prevented the development of oxaliplatin- and paclitaxel-induced mechanical allodynia and hyperalgesia, although the difference was more significant than that in the vehicle group in a few days (Figure 1B–D and Figure 2B–D). In addition, treatment with 100 mg/kg gabapentin significantly prevented the development of oxaliplatin- and paclitaxel-induced mechanical allodynia and hyperalgesia compared with the combination treatment with 30 mg/kg gabapentin and oxaliplatin or paclitaxel (Figure 1B–D and Figure 2B–D). Although treatment of mice with 6 mg/kg oxaliplatin, 6 mg/kg oxaliplatin plus 30 mg/kg gabapentin, 6 mg/kg oxaliplatin plus 100 mg/kg gabapentin, and 6 mg/kg paclitaxel resulted in weight loss in a few days, no dramatic changes were observed (Appendix A). Moreover, no significant differences in weight gain were observed at any time point in the gabapentin group compared with the vehicle group (Appendix A). These observations indicated that gabapentin prevented oxaliplatin- and paclitaxel-induced neuropathic-like pain behavior in a dose-dependent manner.

### 2.2. Effects of Duloxetine on Oxaliplatin- and Paclitaxel-Induced Cold and Mechanical Allodynia

To investigate the effect of duloxetine on oxaliplatin- and paclitaxel-induced neuropathic-like pain behavior, mice received oxaliplatin or paclitaxel on days 0 and 7 and were administered duloxetine (10 or 30 mg/kg/day, p.o.) daily. Oral administration of duloxetine prevented the development of oxaliplatin- and paclitaxel-induced cold allodynia (Figure 3A and Figure 4A). No significant differences were observed in withdrawal latency at any time point by the combination treatment with 10 mg/kg duloxetine, 6 mg/kg oxaliplatin, or 6 mg/kg paclitaxel compared with the combination treatment with 30 mg/kg duloxetine and 6 mg/kg oxaliplatin or 6 mg/kg paclitaxel (Figure 3A and Figure 4A). In addition, we investigated the effect of duloxetine on the development of oxaliplatin- and paclitaxel-induced mechanical allodynia. Administration of 10 mg/kg duloxetine prevented the development of oxaliplatin- and paclitaxel-induced mechanical allodynia and hyperalgesia, although the difference was more significant than that in the vehicle group in a few days (Figure 3B–D and Figure 4B–D). In addition, treatment with 30 mg/kg duloxetine significantly prevented the development of oxaliplatin- and paclitaxel-induced mechanical allodynia and hyperalgesia compared with the combination treatment with 30 mg/kg duloxetine and oxaliplatin or paclitaxel (Figure 3B–D and Figure 4B–D). Although treatment of mice with 6 mg/kg oxaliplatin, 6 mg/kg oxaliplatin plus 10 mg/kg duloxetine, 6 mg/kg oxaliplatin plus 30 mg/kg duloxetine, and 6 mg/kg paclitaxel in mice resulted in a little weight loss in a few days, no dramatic changes were observed (Appendix A). Moreover, no significant differences in weight gain were observed at any time point in the gabapentin group compared with the vehicle group (Appendix A). These observations indicated that duloxetine prevented oxaliplatin- and paclitaxel-induced neuropathic-like pain behavior in a dose-dependent manner.

### 2.3. Gabapentin and Duloxetine Inhibited the Expression of Phosphorylated ERK1/2 in the Lumbar Spinal Cord

We examined whether gabapentin and duloxetine altered the expression of phosphorylated ERK1/2 in the lumbar spinal cord (lumbar segments 4–6) using western blotting. A marked increase in the expression of phosphorylated ERK1/2 was observed in mice treated with oxaliplatin or paclitaxel. Mice treated with gabapentin in addition to oxaliplatin or paclitaxel exhibited significantly reduced phosphorylated ERK1/2 expression compared to those treated with oxaliplatin or paclitaxel alone (Figure 5). Similarly, mice treated with duloxetine in addition to oxaliplatin or paclitaxel exhibited significantly reduced phosphorylated ERK1/2 expression compared to those treated with oxaliplatin or paclitaxel alone. These results indicate that gabapentin and duloxetine suppress oxaliplatin- and paclitaxel-induced neuropathy by inhibiting the expression of phosphorylated ERK1/2.

### 2.4. PD0325901 Prevented Oxaliplatin- and Paclitaxel-Induced Neuropathic-Like Pain Behavior

Our results indicated that gabapentin and duloxetine prevented oxaliplatin- and paclitaxel-induced neuropathic-like pain behavior by suppressing ERK1/2 activation in the lumbar spinal cord (lumbar segments 4–6) of mice. Therefore, we investigated whether PD0325901, a MEK1/2 inhibitor, prevented oxaliplatin- and paclitaxel-induced neuropathy in mice. Treatment with 30 mg/kg PD032501 prevented the development of oxaliplatin- and paclitaxel-induced cold allodynia, mechanical allodynia, and hyperalgesia (Figure 6 and Figure 7). In addition, PD0325901 suppressed oxaliplatin- and paclitaxel-induced ERK1/2 activation in the lumbar spinal cord (lumbar segments 4–6) of mice (Figure 8). These observations indicate that inhibition of ERK1/2 activation by a MEK inhibitor correlate with the prevention oxaliplatin- and paclitaxel-induced neuropathic-like pain by pre-treatment with gabapentin, and duloxetine.

## 3. Discussion

In this study, we demonstrated that gabapentin and duloxetine prevented the development of oxaliplatin- and paclitaxel-induced cold and mechanical allodynia. In addition, 100 mg/kg gabapentin and 30 mg/kg duloxetine significantly suppressed the development of oxaliplatin- and paclitaxel-induced neuropathic-like pain behavior. Moreover, gabapentin and duloxetine treatment did not affect body weight when administered alone, and did not have an effect on the transient loss of body weight in mice treated with oxaliplatin or paclitaxel. Western blot analysis showed that mice treated with gabapentin or duloxetine in addition to oxaliplatin or paclitaxel exhibited significantly reduced phosphorylated ERK1/2 expression compared to those treated with oxaliplatin or paclitaxel alone. Furthermore, we observed that PD0325901 prevented oxaliplatin- and paclitaxel-induced neuropathic-like pain behavior by inhibiting ERK1/2. These findings suggested that gabapentin and duloxetine exerted their analgesic effects of suppressing ERK1/2 phosphorylation in the spinal cord. Our previous study showed that trametinib, a MEK inhibitor, inhibited oxaliplatin-, paclitaxel-, vincristine-, and bortezomib-induced neuropathy by repressing chemotherapy-induced ERK1/2 activation in the lumbar spinal cord of mice [25]. Therefore, ERK1/2 phosphorylation appears to be involved in CIPN, and its inhibition by gabapentin and duloxetine effectively prevented oxaliplatin- and paclitaxel-induced neuropathic-like pain behavior.

We found that 6 mg/kg oxaliplatin and paclitaxel induced neuropathy in mice. Oxaliplatin is injected at 85 mg/m^2^ every 2 weeks or 130 mg/m^2^ every 3 weeks in humans, which corresponds to approximately 6 mg/kg in mice [24]. In addition, several studies have shown that treatment with oxaliplatin (6 or 10 mg/kg) induced neuropathy in mice [26,27,28,29]. In the clinical setting, the recommended paclitaxel dose is 210 mg/m^2^. For an adult human body weighing 60 kg, the paclitaxel dose is 5.9 mg/kg [30]. Moreover, the dose required to induce peripheral neuropathy with paclitaxel is 4–10 mg/kg in mice [31,32,33,34,35]. Therefore, we selected a dose of 6 mg/kg oxaliplatin and paclitaxel in mice. In addition, the equivalent dose (mg/kg) of gabapentin or duloxetine for mice was calculated by multiplying human equivalent doses by a factor of 12.3 [36,37]. The resultant therapeutic equivalent doses of duloxetine and gabapentin were found to be 140 and 28 mg/kg, respectively. Thus, 100 mg/kg (round off) of gabapentin or 30 mg/kg (round up) of duloxetine was administered as the maximum dose to mice.

Oxaliplatin administration induces PKCγ up-regulation and PKCγ and PKCε phosphorylation within the thalamus and periaqueductal gray area in rats [10,38]. In mouse model, treatment with paclitaxel induced persistent activation of PKCβII, PKCδ, and PKCε in the DRG and spinal cord [11,39,40,41]. In addition, PKC-induced ERK1/2 phosphorylation has been associated with chronic pain [10,42]. Moreover, ERKs act as regulators of nociceptive sensitivity in various models of inflammatory pain and mechanical and thermal hyperalgesia, and have been used as measurable marker for sensitization in pain studies [43,44,45,46,47,48]. Therefore, inhibiting ERK signaling may be an effective strategy for preventing chemotherapy-induced neuropathy.

A number of studies have suggested that transient receptor potential (TRP) channels are involved in chemotherapy-induced neuropathic pain [49,50]. Treatment with oxaliplatin increases the expression of TRPV1, TRPV4, TRPA1, and TRPM8, and the upregulation of TRPA1 and TRPM8 in mice and rats is associated with oxaliplatin-induced cold allodynia [51,52,53]. Furthermore, TRPV1 and TRPV4 are involved in paclitaxel-induced cold and mechanical allodynia in rats and mice [54,55,56]. In addition, ERK activation in DRG nociceptive neurons promotes TRPV1 expression [57]. The activation of TRP channels activates several signaling pathways such as the nuclear factor-kappa B pathway and mitogen-activated protein kinase pathways, including ERK and p38 pathways [58,59,60]. Thus, this evidence indicates that ERK1/2 and members of the TRP family interact with each other, and together may be involved in the pathophysiology of neuropathic pain.

It has been reported that administration with 40 mg/kg duloxetine improved the chemotherapy (oxaliplatin-based chemotherapy: FOLFOX therapy, paclitaxel, and bortezomib)-induced peripheral neuropathic pain in patients with multiple myeloma, colon, and breast cancer [61]. It has also indicated that treatment with duloxetine for chemotherapy-induced peripheral neuropathy was moderate recommendation in the American Society of Clinical Oncology guidelines [62]. In addition, administration with gabapentin was improved paclitaxel-induced neuropathy symptoms, neuropathic pain, neurologic deficit, and quality of life in ovarian cancer patients [63]. Our results showed that gabapentin and duloxetine prevented the development of oxaliplatin- and paclitaxel-induced neuropathy via inhibition of ERK1/2 activation in spinal cord of mice. Therefore, the present study showed part of the mechanism of action of gabapentin and duloxetine, which has been shown to be useful in clinical trials.

There is one limitation in our study. Although we used male mice to investigate the preventive effects of gabapentin, duloxetine, and PD0325901 on oxaliplatin- and paclitaxel-induced neuropathic-like pain behavior, did not investigated female mice. It has been reported that sphingosine-1-phosphate receptor subtype 1 (S1PR1) antagonists and A_3_ adenosine receptor subtype (A_3_AR) agonists suppressed the oxaliplatin- and paclitaxel-induced neuropathy in male and female rats, but S1PR1 antagonists and A_3_AR agonists only inhibited the bortezomib-induced neuropathy in male rats. [64]. It has also indicated that resolvin D5 inhibited the paclitaxel-induced mechanical allodynia in male mice, but did not affect female mice [65]. These findings suggest that the effectiveness of drugs to suppress pain-like behavior present sex differences. Thus, future studies will examine the preventive effects of gabapentin, duloxetine, and PD0325901 on oxaliplatin- and paclitaxel-induced neuropathic-like pain behavior in female mice.

## 4. Materials and Methods

### 4.1. Animals

Male 5-week-old BALB/c mice were purchased from Shimizu Laboratory Animals (Kyoto, Japan). The mice were housed in standard cages maintained at 25 °C, kept under a 12 h light/12 h dark cycle, and allowed free access to water and food pellets. All animal experiments were approved by the Animal Care and Use Committee of Kindai University (project identification code KAPS-2020-011, 1 April 2020).

### 4.2. Drugs

Oxaliplatin and PD0325901 were purchased from LC Laboratories (Woburn, MA, USA). Paclitaxel was purchased from FUJIFILM Wako (Tokyo, Japan). Gabapentin was purchased from Sigma (St Louis, MO, USA). Duloxetine was purchased from LKT Laboratories, Inc. (St Paul, MN, USA). Oxaliplatin was dissolved in 5% glucose solution. Paclitaxel was dissolved in cremophor/ethanol/saline (1:1:18, Sigma). Gabapentin, duloxetine, and PD0325901 were dissolved in dimethyl sulfoxide (DMSO), then diluted to 0.5% DMSO in phosphate buffered saline.

### 4.3. Oxaliplatin- and Paclitaxel-Induced Neuropathy Models

After measuring the baseline nociceptive threshold, mice (6-week-old) were administered drugs as per the following schedule. Mice were administered oxaliplatin (6 mg/kg) by intravenous injection (i.v.), paclitaxel (6 mg/kg) by intraperitoneal injection (i.p.), 5% glucose solution, or cremophor/ethanol/saline (vehicle) on days 0 and 7 (*n* = 7 for each group). On day 0, 12 h after treatment with oxaliplatin or paclitaxel, mice were administered gabapentin, duloxetine, or PD0325901. Behavioral tests were performed daily from day 0 to 14. Gabapentin (30 or 100 mg/kg, p.o.), duloxetine (10 or 30 mg/kg, p.o.), PD0325901 (30 mg/kg, p.o.), or 0.5% DMSO (vehicle, p.o.) was administered daily from day 0 to 14 (*n* = 7 for each group) after conducting behavioral tests. The investigator was blinded to the experimental conditions while evaluating the antinociceptive effects of gabapentin, duloxetine, or PD0325901 on the behavioral features of mice.

### 4.4. Behavioral Tests

Behavioral assays were performed as described in a previous study [25,66,67]. Sensitivity to cold was assessed using a hot/cold plate analgesiometer (Ugo Basile, Milan, Italy). Each mouse was placed at the center of a plate maintained at 10 °C (cold allodynia), following which oxaliplatin- and paclitaxel-induced pain-related behaviors, such as lifting and licking of the hind paw, were observed, and the time was recorded (cut-off time was 30 s).

Mechanical allodynia and hyperalgesia were studied using 0.16, 0.4, and 1.4 g of von Frey filaments (Ugo Basile). For each filament, five stimuli were applied at an interval of 3–5 s, and mechanical sensitivity was scored as follows: 0, no response; 1, paw withdrawal; and 2, immediate flinching of the stimulated paw. Aversive response score of five trials from both hind paws of each mouse were averaged and recorded as mean ± S.D.

### 4.5. Western Blotting

Mice were sacrificed, and the lumbar spinal cords were quickly dissected. The dissected tissue was homogenized in ice-cold buffer (20 mM Tris-HCl (Sigma, pH 7.4), 2% Triton X-100 (Sigma), 150 mM NaCl (FUJIFILM Wako), 1 mM EDTA (FUJIFILM Wako), 5 mM MgCl_2_ (FUJIFILM Wako), 10% anhydrous glycerol (FUJIFILM Wako), protease and phosphatase inhibitor cocktail (Roche, Indianapolis, IN, USA)) and centrifuged. The extracts (20 µg of protein) were separated by sodium dodecyl sulfate–polyacrylamide gel electrophoresis and transferred onto polyvinylidene fluoride membranes (GE Healthcare, Buckinghamshire, UK). The membranes were blocked with a solution containing 3% skimmed milk and incubated overnight at 4 °C with anti-phospho-ERK1/2 (Thr202/Tyr204; 1:3000; #9101) (Cell Signaling Technology, Beverly, MA, USA), anti-ERK1/2 (1:3000; #9102, Cell Signaling Technology), and anti-β-actin (1:3000; AC-74, Sigma) antibodies. The membranes were incubated with horseradish peroxidase-coupled anti-rabbit IgG sheep antibodies (GE Healthcare) for 1 h at room temperature. The reactive proteins were visualized using Luminata Forte HRP substrate (Merck Millipore, Nottingham, UK) according to the manufacturer’s instructions.

### 4.6. Statistics

All results are expressed as the mean ± S.D. of at least 4 number of independent experiments. Following ANOVA test, multiple comparisons were conducted using the Tukey test, and the control group and various drug-treated groups were compared and analyzed using Dunnet’s test. Data were tested for normality using the Shapiro-Wilk test. When data were not normally distributed, these were analyzed using the Kruskal-Wallis test followed by the Scheffe test. *p* values < 0.05 were considered significant.

## 5. Conclusions

In summary, our study suggests that the analgesic effects of gabapentin, duloxetine, and PD0325901 against the development of oxaliplatin- and paclitaxel-induced neuropathic-like pain behavior in mice are mediated via the inhibition of ERK1/2 activation. In addition, we demonstrated that the activation of ERK1/2 in the spinal cord plays a critical role in the induction of mechanical and cold allodynia in oxaliplatin- and paclitaxel-induced neuropathic-like pain. Furthermore, these data implicate that inhibiting ERK1/2 activation using gabapentin, duloxetine, PD0325901, or other therapeutic agents could be an effective preventive strategy against oxaliplatin- and paclitaxel-induced neuropathy.

## Figures and Tables

**Figure 1 pharmaceuticals-14-00030-f001:**
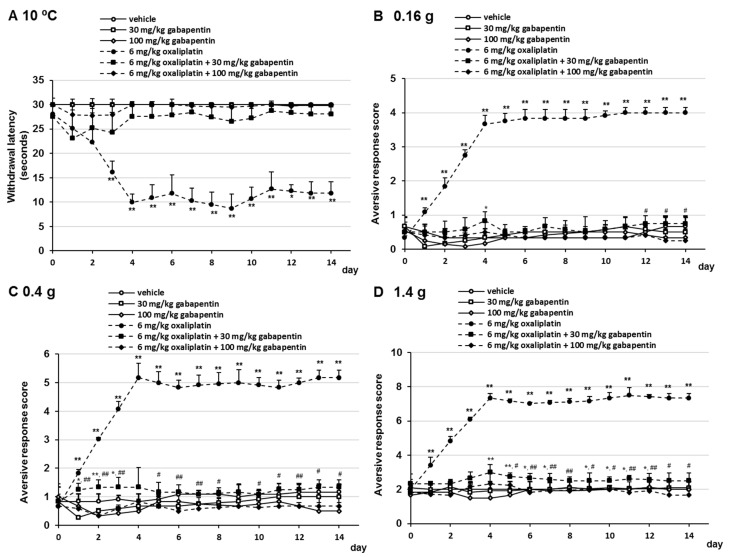
Inhibitory effect of gabapentin on oxaliplatin-induced cold allodynia in mice. Oxaliplatin (6 mg/kg/i.v.) was administered on days 0 and 7, and gabapentin (30 or 100 mg/kg, p.o.) was administered daily from days 0 to 14. (**A**) Withdrawal latencies, presented as means ± standard deviations (S.D.), represent the time taken by mice to withdraw their hind paws following cold stimulation (10 °C) (*n* = 7 per group). * *p* < 0.05, ** *p* < 0.01 vs. vehicles (Shapiro–Wilk test and Kruskal–Wallis test followed by the Scheffe test). (**B**–**D**) The number of paw lifts elicited by five mechanical stimulations using von Frey filaments corresponding to (**B**) innocuous (0.16 g), (**C**) intermediate (0.4 g), and (**D**) noxious (1.4 g) bending forces. The aversive response score was calculated based on two paw lifts (*n* = 7 per group). * *p* < 0.05, ** *p* < 0.01 vs. vehicles, ^#^
*p* < 0.05, ^##^
*p* < 0.01 vs. 6 mg/kg oxaliplatin + 30 mg/kg gabapentin (Shapiro–Wilk test and one-way analysis of variance (ANOVA) with the Tukey test).

**Figure 2 pharmaceuticals-14-00030-f002:**
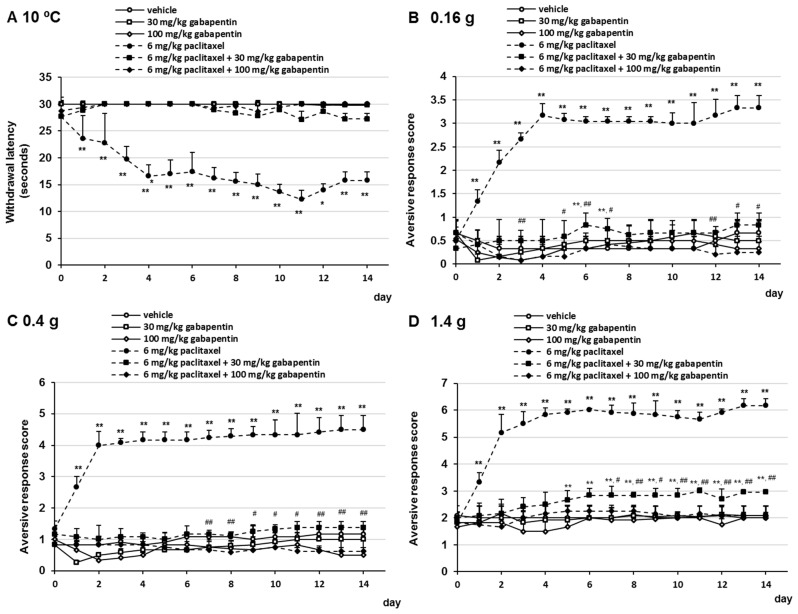
Inhibitory effect of gabapentin on paclitaxel-induced mechanical allodynia in mice. Paclitaxel (6 mg/kg, i.p.) was administered on days 0 and 7, and gabapentin (30 or 100 mg/kg, p.o.) was administered daily from days 0 to 14. (**A**) Withdrawal latencies, presented as means ± standard deviations (S.D.), represent the time taken by mice to withdraw their hind paws following cold stimulation (10 °C) (*n* = 7 per group). * *p* < 0.05, ** *p* < 0.01 vs. vehicles (Shapiro–Wilk test and Kruskal–Wallis test followed by the Scheffe test). (**B**–**D**) The number of paw lifts elicited by five mechanical stimulations using von Frey filaments corresponding to (**B**) innocuous (0.16 g), (**C**) intermediate (0.4 g), and (**D**) noxious (1.4 g) bending forces. The aversive response score was calculated based on two paw lifts (*n* = 7 per group). * *p* < 0.05, ** *p* < 0.01 vs. vehicles, ^#^
*p* < 0.05, ^##^
*p* < 0.01 vs. 6 mg/kg paclitaxel + 30 mg/kg gabapentin (Shapiro–Wilk test and one-way analysis of variance (ANOVA) with the Tukey test).

**Figure 3 pharmaceuticals-14-00030-f003:**
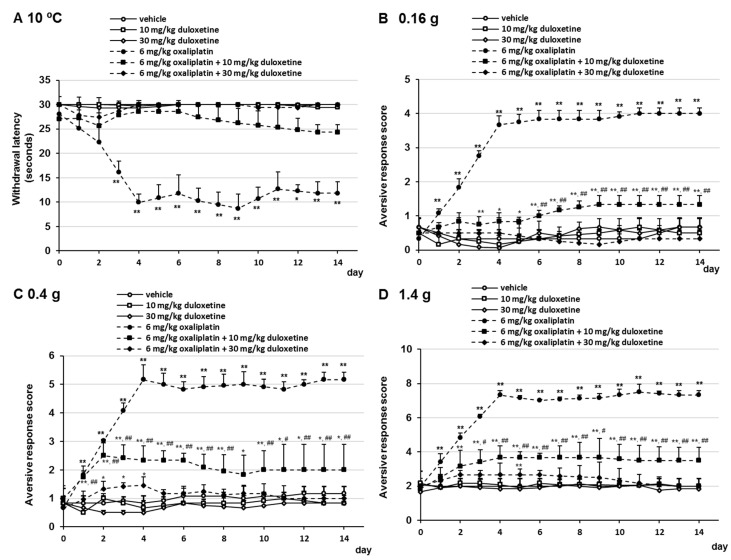
Inhibitory effect of duloxetine on oxaliplatin-induced cold and mechanical allodynia in mice. Oxaliplatin (6 mg/kg, i.v.) was administered on days 0 and 7, and duloxetine (10 or 30mg/kg, p.o.) was administered daily from days 0 to 14. (**A**) Withdrawal latencies, presented as means ± standard deviations (S.D.), represent the time taken by mice to withdraw their hind paws following cold stimulation (10 °C) (*n* = 7 per group). * *p* < 0.05, ** *p* < 0.01 vs. vehicles (Shapiro–Wilk test and Kruskal–Wallis test followed by the Scheffe test). (B–D) The number of paw lifts elicited by five mechanical stimulations using von Frey filaments corresponding to (**B**) innocuous (0.16 g), (**C**) intermediate (0.4 g), and (**D**) noxious (1.4 g) bending forces. The aversive response score was calculated based on two paw lifts (*n* = 7 per group). * *p* < 0.05, ** *p* < 0.01 vs. vehicles, ^#^
*p* < 0.05, ^##^
*p* < 0.01 vs. 6 mg/kg oxaliplatin + 10 mg/kg duloxetine (Shapiro–Wilk test and one-way analysis of variance [ANOVA] with the Tukey test).

**Figure 4 pharmaceuticals-14-00030-f004:**
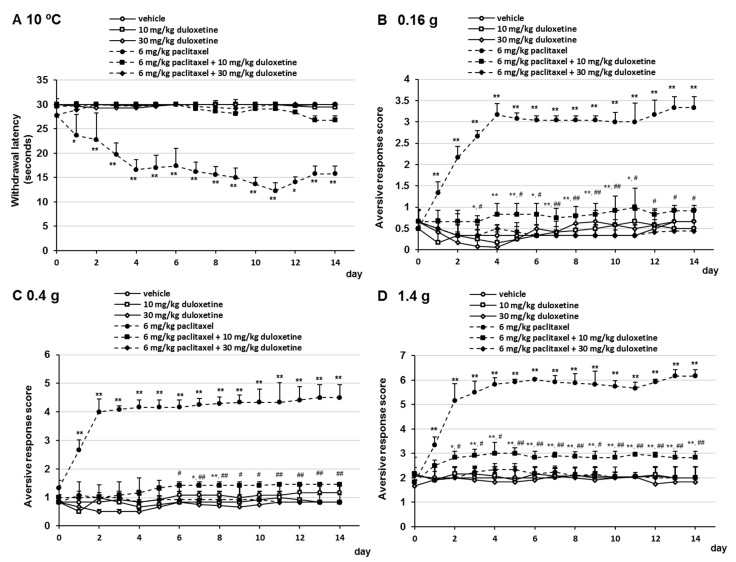
Inhibitory effect of duloxetine on paclitaxel-induced cold and mechanical allodynia in mice. Paclitaxel (6 mg/kg, i.p.) was administered on days 0 and 7, and duloxetine (10 or 30mg/kg, p.o.) was administered daily from days 0 to 14. (**A**) Withdrawal latencies, presented as means ± standard deviations (S.D.), represent the time taken by mice to withdraw their hind paws following cold stimulation (10 °C) (*n* = 7 per group). * *p* < 0.05, ** *p* < 0.01 vs. vehicles (Shapiro–Wilk test and Kruskal–Wallis test followed by the Scheffe test). (**B**–**D**) The number of paw lifts elicited by five mechanical stimulations using von Frey filaments corresponding to (**B**) innocuous (0.16 g), (**C**) intermediate (0.4 g), and (**D**) noxious (1.4 g) bending forces. The aversive response score calculated based on two paw lifts (*n* = 7 per group). * *p* < 0.05, ** *p* < 0.01 vs. vehicles, ^#^
*p* < 0.05, ^##^
*p* < 0.01 vs. 6 mg/kg paclitaxel + 10 mg/kg duloxetine (Shapiro–Wilk test and one-way analysis of variance (ANOVA) with the Tukey test).

**Figure 5 pharmaceuticals-14-00030-f005:**
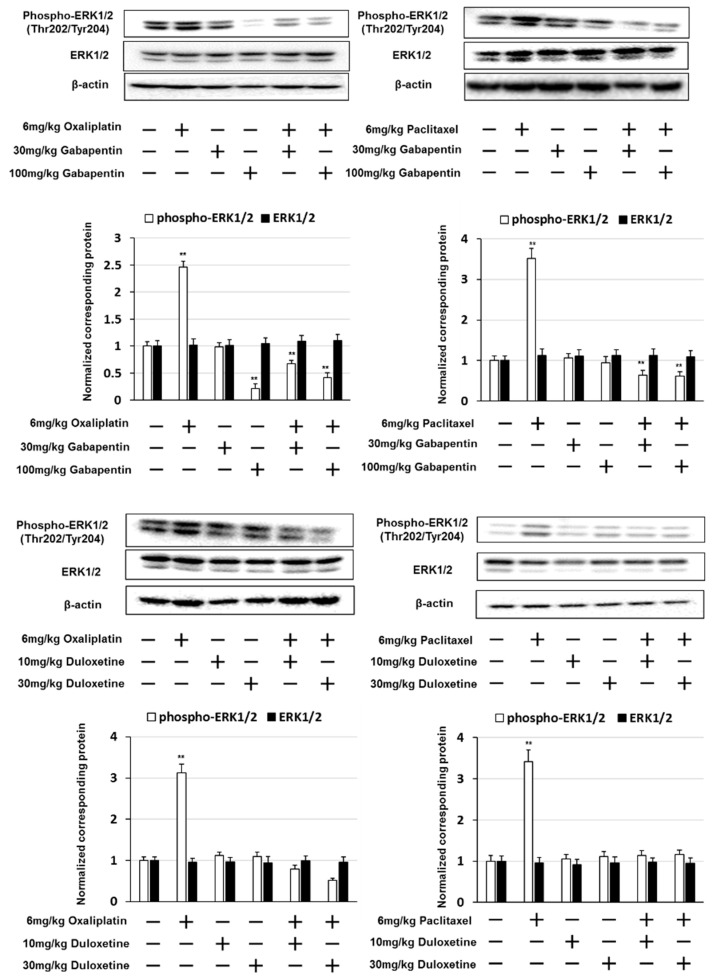
Gabapentin and duloxetine inhibited oxaliplatin- and paclitaxel-induced extracellular signal-regulated kinase 1/2 (ERK1/2) activation. Western blotting was conducted to analyze phosphorylated ERK1/2 (phospho-ERK1/2) protein expression in the spinal cord (L4–L6) obtained from mice on day 14 after treatment with oxaliplatin, paclitaxel, gabapentin, or duloxetine. β-actin was used as a protein loading control. Quantification of phospho-ERK1/2 and ERK1/2 expression, normalized against that of β-actin. The results show four independent experiments. ** *p* < 0.01 vs. controls (Shapiro–Wilk test and ANOVA with Dunnett’s test).

**Figure 6 pharmaceuticals-14-00030-f006:**
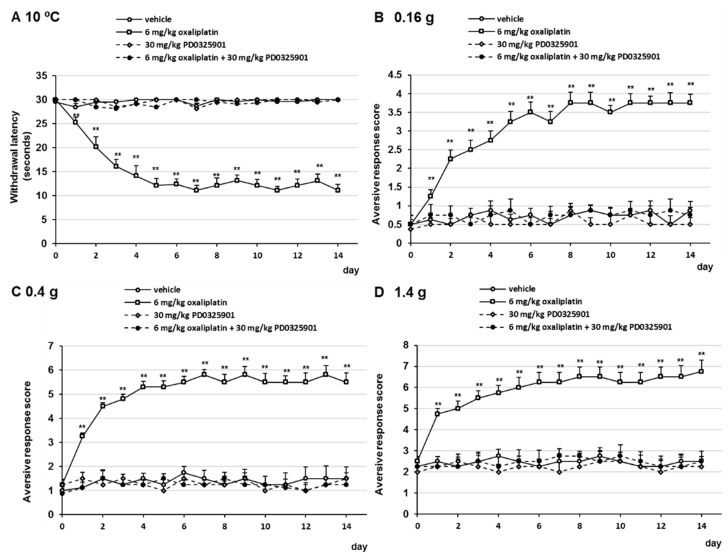
Inhibitory effect of PD0325901 on oxaliplatin-induced cold and mechanical allodynia in mice. Oxaliplatin (6 mg/kg, i.v.) was administered on days 0 and 7, and PD0325901 (30 mg/kg, p.o.) was administered daily from days 0 to 14. (**A**) Withdrawal latencies, presented as means ± standard deviations (S.D.), represent the time taken by mice to withdraw their hind paws following cold stimulation (10 °C) (*n* = 7 per group). ** *p* < 0.01 vs. vehicles (Shapiro–Wilk test and Kruskal–Wallis test followed by the Scheffe test). (**B**–**D**) The number of paw lifts elicited by five mechanical stimulations using von Frey filaments corresponding to (**B**) innocuous (0.16 g), (**C**) intermediate (0.4 g), and (**D**) noxious (1.4 g) bending forces. The aversive response score was calculated based on two paw lifts (*n* = 7 per group). ** *p* < 0.01 vs. vehicles (Shapiro-Wilk test and one-way analysis of variance (ANOVA) with Tukey test).

**Figure 7 pharmaceuticals-14-00030-f007:**
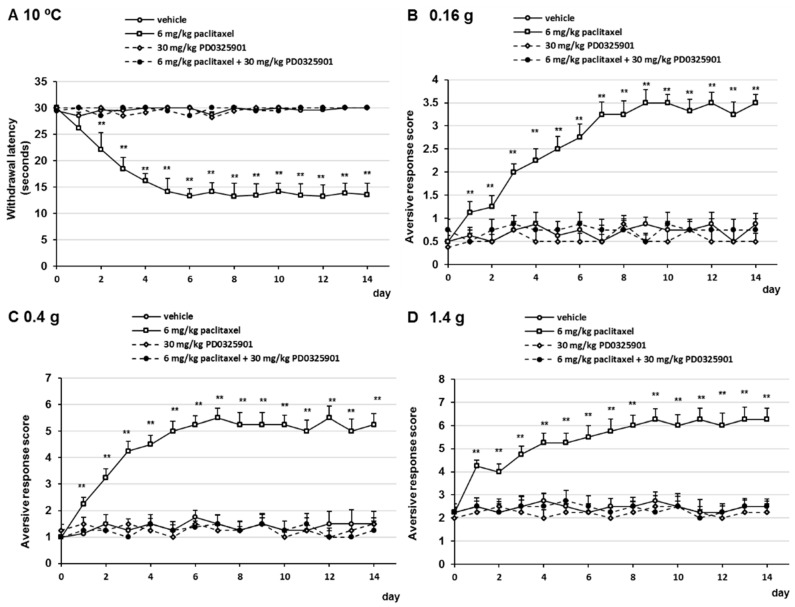
Inhibitory effect of PD0325901 on paclitaxel-induced cold and mechanical allodynia in mice. Paclitaxel (6 mg/kg, i.p.) was administered on days 0 and 7, and PD0325901 (30 mg/kg, p.o.) was administered daily from days 0 to 14. (**A**) Withdrawal latencies, presented as means ± standard deviations (S.D.), represent the time taken by mice to withdraw their hind paws following cold stimulation (10 °C) (*n* = 7 per group). ** *p* < 0.01 vs. vehicles (Shapiro–Wilk test and Kruskal–Wallis test followed by the Scheffe test). (**B**–**D**) The number of paw lifts elicited by five mechanical stimulations using von Frey filaments corresponding to (**B**) innocuous (0.16 g), (**C**) intermediate (0.4 g), and (**D**) noxious (1.4 g) bending forces. The aversive response score was calculated based on two paw lifts (*n* = 7 per group). ** *p* < 0.01 vs. vehicles (Shapiro-Wilk test and one-way analysis of variance (ANOVA) with Tukey test).

**Figure 8 pharmaceuticals-14-00030-f008:**
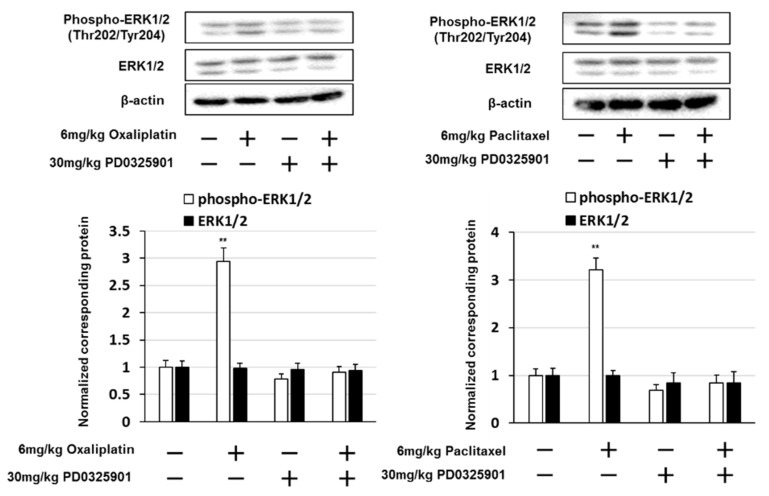
PD0325901 inhibited oxaliplatin- and paclitaxel-induced extracellular signal-regulated kinase 1/2 (ERK1/2) activation. Western blotting was conducted to analyze phosphorylated ERK1/2 (phospho-ERK1/2) protein expression in the spinal cord (L4–L6) obtained from mice on day 14 after treatment with oxaliplatin, paclitaxel, or PD0325901. β-actin was used as a protein loading control. Quantification of phospho-ERK1/2 and ERK1/2 expression, normalized against that of β-actin. The results show four independent experiments. ** *p* < 0.01 vs. controls (Shapiro–Wilk test and ANOVA with Dunnett’s test).

## Data Availability

The data presented in this study are available in the main text and the Appendix A.

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
