# Peer review of "Gabapentin and Duloxetine Prevent Oxaliplatin- and Paclitaxel-Induced Peripheral Neuropathy by Inhibiting Extracellular Signal-Regulated Kinase 1/2 (ERK1/2) Phosphorylation in Spinal Cords of Mice"

_pharmaceuticals, 2020, doi:10.3390/ph14010030_

Round 1
Reviewer 1 Report
The manuscript needs some major corrects:
Material and methods:
- propose to describe more dateiled the timecourse of neuropathy induced by chemotheraputics
- Propose to insert into manuscript the description of behavioral test - it is very unconvenient to the readers to look for it in other referenced manuscirpt
- propose to discuss more clearly why Authors choose the doses of duloxetine and gabapentin . The dose of duloxetine used in the study is much higher than that one used in clinical practice (in clinical practice 120 mg in adult patient, ie around up to 1,7 mg/kg)
Results :
- results obtained in the study indicate that gabapentin and duloxetine PREVENTED the development of neuropathy (gabapentin and duloxetine according to the manuscript were given from day 0, in parallel with chemotheraputics) not alleviated the pain behavior, because it didn't develop - propose to analyze the data, rearrange the results and discussion as well
- propose to analyze the lack of differences between groups: OXA or PAC+ high dose of GABA, OXA or PAC + low dose GABA, it reffers as well to groups treated with duloxetine . I would expect some differences between groups with high dose and low dose of active treatements. I would expect as well some differences between groups GABA or Dul vs GABA or DUL with chemothepautics. Please, discuss it more detailed.
Discussion:
- propose to rearrange discussion after analyzing the results indicating the prevention of neuropathy , not alleviating
- propose to discuss more clearly the doses of adjuvants used in the study
- propose to update the references - 38 of 50 is older than 5 years
- please, refer as well to human studies
After major revisions the manuscript may be published.
Reviewer 2 Report
Manuscript: pharmaceuticals-995633
Title: Gabapentin and duloxetine suppress oxaliplatin- and paclitaxel-induced peripheral neuropathy via inhibition of extracellular signal-regulated kinase 1/2 phosphorylation in the spinal cord of mice.
Review
The manuscript by Kato and colleagues attempts to address an important and clinically-relevant question concerning the pharmacological alleviation of neuropathic side-effects of chemotherapeutic agents.
The authors present a series of sensory behaviour experiments in mice examining the effect of two chemotherapies (systemic paclitaxel or oxaliplatin administration) on withdrawal responses to cold and mechanical (von Frey) stimulation of the hind paw – a surrogate marker of neuropathic pain. The paper presents data on the effects of two doses of daily pharmacological treatment with either gabapentin or duloxetine, two drugs already used in the clinic for painful peripheral neuropathies, including chemotherapy-induced neuropathy. The authors also perform western blot for phosphorylated ERK1/2 on proteins extracted from spinal cords of mice 14 days after paclitaxel or oxaliplatin treatment.
My two main concerns with this study is the lack of novelty, and lack of experiment detail for the reader to assess the quality of the data presented.
Firstly, the effect of gabapentin and duloxetine on both cold and mechanical pain induced by oxaliplatin and paclitaxel is presented as a novel finding (see abstract), when by the authors' own admission this has been shown before (refs 22-24).
A brief literature search in fact reveals a host of similar studies examining the effect of gabapentin and duloxetine on both cold and mechanical pain in rodent models of oxaliplatin and paclitaxel induced neuropathy. Effect of gabapentin on oxaliplatin–induced cold allodynia: PMID: 24714523, PMID: 24671055. Effect of duloxetine on oxaliplatin-induced cold sensitivity: PMID: 24525711. Effect of repeat dosing of gabapentin on paclitaxel-induced mechanical allodynia: PMID: 17084535. Effect of duloxetine on paclitaxel-induced mechanical allodynia: PMID: 22460833.
I appreciate the need to confirm the model in the hands of the authors, but I question why were so many animal experiments (multiple doses and drug combinations) were necessary to achieve this, especially considering the effect of either drug at either dose is apparently so obvious. Furthermore, nowhere in the paper does it state the numbers of animals used in each experiment. Sadly, female mice are omitted from the study.
Secondly, no effort has been made to quantify the western blot experiments. Presumably each lane in each blot represents an individual animal. But how many times was each experiment performed?
Lastly, the data together offer no more than circumstantial evidence for either gabapentin or duloxetine suppressing neuropathic pain in chemotherapy by reducing ERK phosphorylation. Considering the importance of this signalling pathway in numerous biological processes, I’m not convinced it offers a more advantageous pharmacological target over and above the use of gabapentin and duloxetine.
Pharmacologically targeting ERK phosphorylation directly would lend at least some support to the authors’ hypothesis.
Specific comments:
- The authors must state the numbers of animals used, both in each experiment and within the study overall. The statistics are meaningless without this information.
- Please include a description of the behaviour tests. It is not acceptable to expect the reader to search the previous literature for this information. For example, how many times was each paw stimulated? I.e. what is the percentage response?
- How many cohorts of mice were tested? Were vehicle and drug-treated mice examined at the same time? Was the investigator blind to the treatment group? What age were the mice at the start of the experiment?
- Mechanical withdrawal responses were assessed with different grades of von Frey filament. Wouldn’t application of the ‘noxious’ filament (1.4g) be expected to increase the number of responses at baseline above that seen with the innocuous (0.16g) filament?
- Gabapentin is sedative. How did the authors control for this potential confound when assessing withdrawal behaviours after treatment?
- The figure legends report p-values obtained by ANOVA. Exactly what comparisons were made? If treatment groups were compared over time then a two-way ANOVA is appropriate. Please include the F statistic, with degrees of freedom, and state what comparisons are being reported (e.g. versus baseline, or versus vehicle).
- The western blots need quantification and clear details on the numbers of experiments performed. By eye it is impossible to assess whether even oxaliplatin has an effect on p-ERK levels.
- In supplementary data, the authors show the body weight of mice during each experiment. It is surprising that no mice increase their body weight above 30 g for the duration of the study. It may be preferable to show % body weight. An apparent decrease in weight from 30g to 27g (10% loss) is potentially significant.
- Both gabapentin and duloxetine are made up in DMSO and administered orally (p.o.). What was the vehicle and final concentration of DMSO? Was this given in drinking water, or as a gavage?
- The images of the full blots presented in supplementary data is welcome, although I cannot see a description of this figure in the paper. I see a semblance of these blots in the main Fig 5. Have the images been re-sized or undergone any other kind of processing?
Minor
- I recommend annotating each behavioural data graph with the experimental conditions to make it easier for reader to see the difference. E.g. ‘Cold’, ‘0.16 g’, ‘0.4g’ etc.
- What concentrations of antibodies were used for the western blots?
- What is the composition of the tissue lysis buffer for spinal cord homogenisation?
Reviewer 3 Report
The novelty of these findings is very low, the title and abstract overstate results. It is not demonstrated that the pain relieving effects are "via" ERK1/2, data show only a modulation of their expression
Round 2
Reviewer 1 Report
Dear Authors,
I'm very glad to see the manuscript corrected. In my opinion the correction you've made increase the clarity and scientific importance of your paper.
I accept your answers and corrections.
Reviewer 2 Report
The manuscript has been improved with new data using an inhibitor of ERK phosphorylation to show the possible mechanism via which gabapentin and duloxetine prevent the development of mechanical and cold allodynia during paclitaxel and oxaliplatin treatment in mice. This raises the novelty of their findings. The authors also add critical experimental details, including quantification of Western blot data, to facilitate reproduction of their results by others.
There are still some important issues that the authors must address before the study is published:
- The supplementary data recording the body weight are now on the order of 20g, whereas in the previous submitted version the average weight the mice was around 30g. Has there been an error introduced in the data somewhere? Or have these experiments all been repeated with younger mice?
- The updated method details describe scoring of the response to von Frey filaments (0, 1 or 2). However the axes of the graphs indicate the ‘number of withdrawal responses’. Which is correct?
- Please describe how gabapentin, duloxetine, or the PD compound were given orally and add the final composition of the vehicle (0.5% DMSO in tap water?).
- Line 183: “These observations indicated that inhibition of ERK1/2 activation by MEK inhibitor, 183 gabapentin, and duloxetine prevented oxaliplatin- and paclitaxel-induced neuropathy.” This conclusion is too strong. Please change to the following: ‘These observations indicate that inhibition of ERK1/2 activation by a MEK inhibitor correlate with the prevention oxaliplatin- and paclitaxel-induced neuropathic-like pain by pre-treatment with gabapentin, and duloxetine.’
- The study only looks one aspect of chemotherapy-induced peripheral neuropathy – namely mechanical and cold sensitivity. In mice, these measurements are only a surrogate of neuropathic pain. Therefore, please refer to ‘neuropathic-like pain behaviour’ instead of ‘neuropathy’. For example, Line 29: Please change to: ‘Moreover, PD0325901 prevented the development of oxaliplatin- and paclitaxel-induced neuropathic-like pain behaviour in mice.’
- Line 335: Please change to: ‘In summary, our study suggests that the analgesic effects of gabapentin, duloxetine, and PD0325901 against the development of oxaliplatin- and paclitaxel-induced neuropathic-like pain behaviour in mice are mediated via the inhibition of ERK1/2 activation.’
- Please make the following changes to 4.6 Statistics:
Line 327: All results are expressed as the mean ± S.D. of at least x number of independent experiments.
Line 327: Following ANOVA test, multiple comparisons were conducted using the Tukey test, and the control group and various drug-treated groups were compared and analyzed using Dunnet’s test.
Line 330: ‘Data were tested for normality using the Shapiro-Wilk test. When data were not normally distributed, these were analyzed using Kruskal-Wallis test followed by Scheffe test.’
- Please state clearly the n number next to each statistical result.
- It is not sufficient to reference other similar studies that did not use female animals. The lack of data using female animals must be stated as limitation to the translatability of the current study. Please add this statement to the discussion.
- Please add a legend to the images of the full blots in the supplementary Figures.
